# The Differences Between Direct Alignment Algorithms are a Blur

**Alexey Gorbatovski, Boris Shaposhnikov,**\* **Viacheslav Sinii, Alexey Malakhov, Daniil Gavrilov**
T-Tech

## Abstract

Direct Alignment Algorithms (DAAs) simplify language model alignment by replacing reinforcement learning (RL) and reward modeling (RM) in Reinforcement Learning from Human Feedback (RLHF) with direct policy optimization. DAAs can be classified by their ranking losses (pairwise vs. pointwise), by the rewards used in those losses (e.g., likelihood ratios of policy and reference policy, or odds ratios), or by whether a Supervised Fine-Tuning (SFT) phase is required (two-stage vs. one-stage). We first show that one-stage methods underperform two-stage methods. To address this, we incorporate an explicit SFT phase and introduce the $\beta$ parameter, controlling the strength of preference optimization, into single-stage ORPO and ASFT. These modifications improve their performance in Alpaca Eval 2 by +3.46 (ORPO) and +8.27 (ASFT), matching two-stage methods like DPO. Further analysis reveals that the key factor is whether the approach uses pairwise or pointwise objectives, rather than the specific implicit reward or loss function. These results highlight the importance of careful evaluation to avoid premature claims of performance gains or overall superiority in alignment algorithms.

## 1 Introduction

Large Language Models (LLMs) demonstrate strong text generation capabilities, yet aligning them with human values remains challenging due to underspecified objectives, limited training signals, and the complexity of human intent (Ouyang et al., 2022; Stiennon et al., 2020). Traditional alignment pipelines typically involve Supervised Fine-Tuning (SFT), reward modeling, and reinforcement learning to shape model outputs.

Recently, Direct Alignment Algorithms (DAAs) have emerged as an alternative, integrating human preferences into policy optimization without explicit reward modeling or reinforcement learning (Rafailov et al., 2023; Hong et al., 2024; Azar et al., 2023; Meng et al., 2024; Chen et al., 2024; Xiao et al., 2024; D'Oosterlinck et al., 2024; Wang et al., 2024). These methods differ in theoretical design (pairwise vs. pointwise), implementation details (e.g., reference policy vs. odds ratio), and whether an SFT phase is required (one-stage vs. two-stage). This diversity raises key questions about their relationships, comparative advantages, and the role of SFT.

In this paper, we show that one-stage methods (e.g., ORPO, ASFT) can incorporate an explicit SFT phase, improving performance. We introduce a scaling parameter $\beta$ that unifies their formulation with other DAAs, revealing shared optimization dynamics between methods using either an odds ratio or a reference-based reward. Through theoretical and empirical analysis, we systematically compare DAAs, emphasizing pairwise vs. pointwise preference optimization. We also show that, while SFT is beneficial, using the full dataset is not always necessary, which reduces computational costs. To structure our analysis, we address the following research questions:

**RQ1:** Does an explicit SFT stage improve the alignment quality of ORPO and ASFT?

**RQ2:** Does the tempering factor enhance the alignment quality of ASFT and ORPO?

**RQ3:** What factors of DAAs affect alignment quality?

---

\*Correspondence to: Boris Shaposhnikov - b.shaposhnikov@tbank.ru

**RQ4:** How does the final alignment quality depend on the amount of data used in the SFT stage?

By answering these questions, we clarify key trade-offs in alignment strategies and provide guidance for optimizing LLM training pipelines.

## 2 Preliminaries

### 2.1 Modeling Sequences

Given a sequence $y$ of length $|y|$, the log-probability can be written as $\log p(y) = \sum_{i=1}^{|y|} \log p(y_i \mid y_{<i})$, which may also be conditioned on another sequence $x$. In practice, optimizing normalized log-probability $\frac{1}{|y|} \log p(y) = \log\left(p(y)^{\frac{1}{|y|}}\right)$ often improves numerical stability and leads to better training. However, once normalized, the resulting quantity is no longer a strict probability measure. Throughout this paper, whenever we write $p(y)$, we refer to this normalized version $p(y)^{\frac{1}{|y|}}$. Whenever a method does not apply this normalization, we indicate it explicitly.

Welleck et al. (2019) introduced a log-unlikelihood term that reduces the probability of certain undesirable tokens: $\log\left(1 - p(c \mid y_{<i})\right)$ for $c \in \mathcal{C}$. It can be extended to an entire sequence as $\log\left(1 - p(y)\right)$.

### 2.2 Direct Alignment Algorithms

Direct alignment algorithms replace the reward modeling and RL stages (more details in Appendix A) (but keep the SFT phase) with a single alignment step. Various preference-optimization loss functions have been proposed, employing these core components:

- $r_\theta^{\mathrm{ref}}(y, x) = \log\left(\frac{\pi_\theta(y|x)}{\pi_{\mathrm{ref}}(y|x)}\right)$ from DPO (Rafailov et al., 2023), which acts as an implicit reward $\beta\, r_\theta^{\mathrm{ref}}$. No length normalization is used.

- $r_\theta^{\mathrm{odds}}(y, x) = \log\left(\frac{\pi_\theta(y|x)}{1-\pi_\theta(y|x)}\right)$ proposed in ORPO (Hong et al., 2024), representing the odds of generating $y$ versus not generating it.

Several Direct Alignment Algorithms use these notations. Information on sequence probability normalization for these methods is presented in Appendix B.1.

- **Direct Preference Optimization (DPO)** (Rafailov et al., 2023): $\mathcal{L}_{\mathrm{DPO}} = -\log \sigma\left(\beta\, r_\theta^{\mathrm{ref}}(y_w, x) - \beta\, r_\theta^{\mathrm{ref}}(y_l, x)\right)$. This method does not normalize probabilities by length.[1]

- **Identity Preference Optimization (IPO)** (Azar et al., 2023): $\mathcal{L}_{\mathrm{IPO}} = \left(r_\theta^{\mathrm{ref}}(y_w, x) - r_\theta^{\mathrm{ref}}(y_l, x) - \frac{1}{2\beta}\right)^2$.

- **Simple Preference Optimization (SimPO)** (Meng et al., 2024): $\mathcal{L}_{\mathrm{SimPO}} = -\log \sigma\left(\beta \log \pi_\theta(y_w, x) - \beta \log \pi_\theta(y_l, x) - \gamma\right)$.

- **Noise Contrastive Alignment (NCA)** (Chen et al., 2024): $\mathcal{L}_{\mathrm{NCA}} = -\log \sigma\left(\beta\, r_\theta^{\mathrm{ref}}(y_w, x)\right) - 0.5 \log \sigma\left(-\beta\, r_\theta^{\mathrm{ref}}(y_w, x)\right) - 0.5 \log \sigma\left(-\beta\, r_\theta^{\mathrm{ref}}(y_l, x)\right)$.

- **Calibrated Direct Preference Optimization (Cal-DPO)** (Xiao et al., 2024): $\mathcal{L}_{\mathrm{Cal-DPO}} = -\log \sigma\left(r_\theta^{\mathrm{ref}}(y_w, x) - r_\theta^{\mathrm{ref}}(y_l, x)\right) + \left(r_\theta^{\mathrm{ref}}(y_w, x) - \frac{1}{2\beta}\right)^2 + \left(r_\theta^{\mathrm{ref}}(y_l, x) + \frac{1}{2\beta}\right)^2$.

- **Anchored Preference Optimization Zero (APO-Zero)** (D'Oosterlinck et al., 2024): $\mathcal{L}_{\mathrm{APO-Zero}} = -\sigma\left(\beta\, r_\theta^{\mathrm{ref}}(y_w, x)\right) + \sigma\left(\beta\, r_\theta^{\mathrm{ref}}(y_l, x)\right)$.

---

[1] Unless otherwise noted, the expectation over $(x, y_w, y_l) \sim \mathcal{D}$ is taken.

## 2.3 Single-Stage Alignment Methods

Single-stage alignment (as a subset of DAA methods) merges SFT and direct alignment in one step by adding their losses: $\mathcal{L}_{\text{Single}}(\pi_\theta) = -\mathbb{E}_{(x,y_w,y_l)\sim\mathcal{D}}\big[\mathcal{L}_{\text{SFT}}(\pi_\theta, x, y_w) + \lambda\,\mathcal{L}_{\text{Align}}(\pi_\theta, x, y_w, y_l)\big]$, where $\lambda$ is a hyperparameter, and no reference policy $\pi_{\text{ref}}$ is required.

In this paper, we focus on:

- **Odds Ratio Preference Optimization (ORPO)** (Hong et al., 2024): $\mathcal{L}_{\text{ORPO}} = -\log\pi_\theta(y_w|x) - \lambda\underbrace{\log\sigma\big(r_\theta^{\text{odds}}(y_w, x) - r_\theta^{\text{odds}}(y_l, x)\big)}_{-\mathcal{L}_{\text{ORPO}_{\text{Align}}}}.$

- **Aligned Supervised Fine-Tuning (ASFT)** (Wang et al., 2024): $\mathcal{L}_{\text{ASFT}} = -\log\pi_\theta(y_w|x) - \lambda\Big(\underbrace{\log\sigma\big(r_\theta^{\text{odds}}(y_w, x)\big) - \log\sigma\big(-r_\theta^{\text{odds}}(y_l, x)\big)}_{-\mathcal{L}_{\text{ASFT}_{\text{Align}}}}\Big).$

## 3 Method

Many DAAs have been proposed, raising questions about their differences and significance. They can be categorized in various ways. For example, one classification separates single-stage methods, which perform alignment directly after obtaining a base model (ASFT and ORPO), from two-stage methods (which perform SFT before alignment), as in DPO, IPO, SimPO, etc. Under this scheme, ASFT and ORPO are single-stage methods.

Another classification considers whether $r^{\text{ref}}$ or $r^{\text{odds}}$ is used as an implicit reward. ASFT and ORPO also differ from other losses by using an odds ratio, whereas other methods in Section 2 use normalized policy probabilities.[2]

DAAs can also be distinguished by whether their loss function is optimized for pairwise or pointwise preferences. DPO, for instance, increases the policy's probability of choosing preferred sequences relative to rejected ones. In contrast, ASFT simply increases or decreases probabilities for chosen or rejected sequences without comparing them directly.

### 3.1 Generalizing ASFT and ORPO

Despite these classifications, it can still be difficult to pinpoint the essential differences among DAAs, especially when design choices limit generalization. ASFT and ORPO, for example, lack a parameter $\beta$, probably because they were conceived as single-stage methods, making the distance from a reference policy unnecessary. It might seem odd to introduce such a parameter in single-stage methods, but we will show that for both ASFT and ORPO, the single-stage design and the absence of $\beta$ are not strictly required.

#### 3.1.1 ORPO and ASFT can operate without the SFT loss term and as two-stage methods.

We begin by inspecting the ASFT objective and demonstrate that it combines both likelihood and unlikelihood terms:

**Theorem 3.1.** *$\mathcal{L}_{\text{ASFT}}$ is equivalent to the Binary Cross-Entropy (BCE) loss, encapsulating both likelihood and unlikelihood components:*

$$\mathcal{L}_{\text{ASFT}} = -(1 + \lambda)\log\pi_\theta(y_w|x) - \lambda\log\big(1 - \pi_\theta(y_l|x)\big).$$

*The proof of Theorem 3.1 is provided in Appendix C.* Consequently,

$$\mathcal{L}_{\text{ASFT}_{\text{Align}}} = -\Big(\log\pi_\theta(y_w|x) + \log\big(1 - \pi_\theta(y_l|x)\big)\Big).$$

---

[2]SimPO does not explicitly use a reference policy, but can be treated similarly if a uniform reference policy is assumed.

Next, we derive a direct relationship between $\mathcal{L}_{\text{ORPO}}$ and $\mathcal{L}_{\text{ASFT}}$, showing that the latter provides an upper bound on the former:

**Theorem 3.2.** $\mathcal{L}_{\text{ORPO}}$ *can be expressed as:*

$$\mathcal{L}_{\text{ORPO}} = \mathcal{L}_{\text{ASFT}} + \lambda \log\big(\pi_\theta(y_w|x)(1 - \pi_\theta(y_l|x)) + \pi_\theta(y_l|x)(1 - \pi_\theta(y_w|x))\big),$$

*where the additional term is symmetric in $y_w$ and $y_l$.*

*The proof of Theorem 3.2 is provided in Appendix D. As for $\mathcal{L}_{\text{ASFT}_{\text{Align}}}$, the alignment term is then*

$$\mathcal{L}_{\text{ORPO}_{\text{Align}}} = -\log \pi_\theta(y_w|x) - \log(1 - \pi_\theta(y_l|x))$$
$$+ \log\big(\pi_\theta(y_w|x)(1 - \pi_\theta(y_l|x)) + \pi_\theta(y_l|x)(1 - \pi_\theta(y_w|x))\big).$$

**Corollary 3.2.1.** $\mathcal{L}_{\text{ORPO}} \leq \mathcal{L}_{\text{ASFT}}$ *and* $\mathcal{L}_{\text{ORPO}_{\text{Align}}} \leq \mathcal{L}_{\text{ASFT}_{\text{Align}}}$.

This follows from the fact that the additional term in $\mathcal{L}_{\text{ORPO}}$ is non-positive when $\pi_\theta(y_w|x)$ and $\pi_\theta(y_l|x)$ lie in $[0, 1]$, and $\pi_\theta(y_w|x) + \pi_\theta(y_l|x) \leq 1$.

These findings yield two main observations:

- $\mathcal{L}_{\text{ASFT}}$ provides an upper bound on $\mathcal{L}_{\text{ORPO}}$. Minimizing the former also minimizes the latter.

- $\mathcal{L}_{\text{ASFT}}$ can be viewed as a minimal form of a DAA loss, reflecting the structure of BCE.

An essential insight from these formulations is that the SFT term in the ASFT and ORPO losses is already included in the full loss. We hypothesize that this feature may allow us to omit the SFT term in the complete loss, first performing an SFT phase and then using only the alignment terms for model alignment. From this perspective, one can experiment with these methods in both single-stage and two-stage configurations to see which approach is more effective.

### 3.1.2 Tempering ASFT and ORPO

We now consider the original single-stage methods from Section 2.3 and examine how the alignment terms $\mathcal{L}_{\text{ORPO}_{\text{Align}}}$ and $\mathcal{L}_{\text{ASFT}_{\text{Align}}}$ compare. These terms optimize preferences and, depending on the coefficient $\lambda$, can dominate or have a smaller impact on the final loss.

$\mathcal{L}_{\text{ASFT}_{\text{Align}}}$ and $\mathcal{L}_{\text{ORPO}_{\text{Align}}}$ strongly resemble the DAA losses discussed in Section 2.2. The single-stage analogue of $r_\theta^{\text{ref}}$ is $r_\theta^{\text{odds}}$. Inspired by this analogy, we introduce a coefficient $\beta$ to scale $r_\theta^{\text{odds}}$:

$$\mathcal{L}_{\text{ASFT}_{\text{Align}}}^\beta = -\log \sigma(\beta r_\theta^{\text{odds}}(y_w, x)) - \log \sigma(-\beta r_\theta^{\text{odds}}(y_l, x)),$$
$$\mathcal{L}_{\text{ORPO}_{\text{Align}}}^\beta = -\log \sigma(\beta r_\theta^{\text{odds}}(y_w, x) - \beta r_\theta^{\text{odds}}(y_l, x)).$$

Both $\mathcal{L}_{\text{ASFT}}^\beta$ and $\mathcal{L}_{\text{ORPO}}^\beta$ generalize their vanilla counterparts (recovering them when $\beta = 1$). As in DPO, $\beta$ can be viewed as a *temperature* or *scaling* parameter that regulates the intensity of the preference for "good" odds. This becomes clearer when looking at the gradients:

$$\nabla_\theta \mathcal{L}_{\text{ASFT}_{\text{Align}}}^\beta = -\beta \Big[\sigma(\beta r_\theta^{\text{odds}}(y_l, x))\nabla_\theta r_\theta^{\text{odds}}(y_l, x) + \big(1 - \sigma(\beta r_\theta^{\text{odds}}(y_w, x))\big)\nabla_\theta r_\theta^{\text{odds}}(y_w, x)\Big],$$

$$\nabla_\theta \mathcal{L}_{\text{ORPO}_{\text{Align}}}^\beta = -\beta \Big[\big(\nabla_\theta r_\theta^{\text{odds}}(y_w, x) - \nabla_\theta r_\theta^{\text{odds}}(y_l, x)\big) \times \big(1 - \sigma(\beta r_\theta^{\text{odds}}(y_w, x) - \beta r_\theta^{\text{odds}}(y_l, x))\big)\Big],$$

where $\nabla_\theta r_\theta^{\text{odds}}(y, x) = \frac{\nabla_\theta \log \pi_\theta(y|x)}{1 - \pi_\theta(y|x)}$. When $\beta \to 0$, $\sigma(\beta \cdots) \approx \frac{1}{2}$, both methods aggressively improve the odds ratio (increasing for $y_w$ and decreasing for $y_l$). As $\beta$ increases, the updates become bounded by the factor $\sigma(\beta \cdots)$ (similar to a reward threshold in DPO). Hence, once the model improves, further updates are limited, either individually for $\mathcal{L}_{\text{ASFT}_{\text{Align}}}^\beta$ or by pairwise ranking in $\mathcal{L}_{\text{ORPO}_{\text{Align}}}^\beta$. This alignment with other DAAs allows for a direct comparison of all methods in different setups, clarifying which aspects are most critical for successful performance.

### 3.2 On the Difference Between Direct Alignment Algorithms

Different methods can be grouped by the type of "reward" function used in their loss. In general terms, $\mathcal{L}_{\text{ASFT}_{\text{Align}}}^\beta$ and $\mathcal{L}_{\text{ORPO}_{\text{Align}}}^\beta$ employ an odds ratio, while DPO, IPO, SimPO, NCA, Cal-DPO, and APO-Zero use a ratio between the probability of the policy and that of a reference policy.

The following theorems make this classification clearer:

**Theorem 3.3.** *The gradient of $\mathcal{L}^{\beta}_{\text{ASFT}_{\text{Align}}}$ becomes collinear with the gradient of $\mathcal{L}_{\text{ORPO}_{\text{Align}}}$ as $\beta \to 0$. Formally,*

$$\lim_{\beta \to 0} \frac{\nabla_\theta \mathcal{L}^{\beta}_{\text{ASFT}_{\text{Align}}}}{\|\nabla_\theta \mathcal{L}^{\beta}_{\text{ASFT}_{\text{Align}}}\|} = \frac{\nabla_\theta \mathcal{L}_{\text{ORPO}_{\text{Align}}}}{\|\nabla_\theta \mathcal{L}_{\text{ORPO}_{\text{Align}}}\|},$$

*indicating that both gradients point in the same direction.*

*The proof of Theorem 3.3 is provided in Appendix E.1.*

A related property applies to $\mathcal{L}^{\beta}_{\text{ORPO}_{\text{Align}}}$:

**Theorem 3.4.** *The gradient of $\mathcal{L}^{\beta}_{\text{ORPO}_{\text{Align}}}$ is collinear with the gradient of $\mathcal{L}_{\text{ORPO}_{\text{Align}}}$ for any $\beta > 0$. Formally,*

$$\frac{\nabla_\theta \mathcal{L}^{\beta}_{\text{ORPO}_{\text{Align}}}}{\|\nabla_\theta \mathcal{L}^{\beta}_{\text{ORPO}_{\text{Align}}}\|} = \frac{\nabla_\theta \mathcal{L}_{\text{ORPO}_{\text{Align}}}}{\|\nabla_\theta \mathcal{L}_{\text{ORPO}_{\text{Align}}}\|}, \quad \beta > 0.$$

*The proof of Theorem 3.4 is provided in Appendix F.1.*

Finally:

**Theorem 3.5.** *For each method $X \in \{\text{IPO, SimPO, NCA, Cal-DPO, APO-Zero}\}$, as $\beta \to 0$, the gradient of $\mathcal{L}_X$ is collinear with the gradient of $\mathcal{L}_{\text{DPO}}$. Formally,*

$$\lim_{\beta \to 0} \frac{\nabla_\theta \mathcal{L}_X}{\|\nabla_\theta \mathcal{L}_X\|} = \frac{\nabla_\theta \mathcal{L}_{\text{DPO}}}{\|\nabla_\theta \mathcal{L}_{\text{DPO}}\|}.$$

*The proof of Theorem 3.5 is provided in Appendix G.1.*

These theorems suggest that for sufficiently small $\beta$, these loss functions are split into two categories with indistinguishable gradient directions. Although the magnitudes may differ and they may not be collinear for $\beta \not\to 0$, one could infer that their performance should be similar when $\beta$ is small. From this perspective, two main distinctions arise among these methods: the use of an odds ratio ($r_\theta^{\text{odds}}$) and the use of the ratio to a reference policy ($r_\theta^{\text{ref}}$). Both choices might influence the final performance of these methods. Furthermore, it remains an open question whether odds-ratio-based approaches outperform reference-policy-based ones (e.g., DPO), and how these distinctions compare to the contrast between pointwise and pairwise preference formulations. From traditional learning-to-rank Liu et al. (2009) research, pairwise methods often produce more direct and less noisy ranking signals than pointwise techniques, which could lead to superior performance in practice (Burges et al., 2005; Li, 2011; Melnikov et al., 2016). In the following sections, we present experimental results that provide further insight into which aspects most strongly influence DAA training.

## 4 EXPERIMENTAL SETUP

We systematically compare and evaluate DAA methods using a standard training and instruction-following evaluation framework Tunstall et al. (2023); Meng et al. (2024); Gorbatovski et al. (2024). Our main experiments use the Llama 3.1 8B model AI@Meta (2024), trained on the UltraChat Ding et al. (2023) and UltraFeedback (UF) Cui et al. (2023) datasets, and evaluated on the AlpacaEval 2 Dubois et al. (2024); Li et al. (2023) and ArenaHard Li et al. (2024) benchmarks. For the Reddit TL;DR Stiennon et al. (2020) task, we employ the Llama 3.2 3B model, comparing it side by side with the "golden" validation split Rafailov et al. (2023; 2024) using the prompt in Appendix K.

### 4.1 BASE VS SFT-INITIALIZED MODELS.

To investigate the impact of SFT and the applicability of one-stage loss $\mathcal{L}_{\text{Align}}$ component, we use the UF dataset for SFT (avoiding additional knowledge from UltraChat), and for pairwise preference optimization. We carefully tuned the hyperparameters to optimize each method's performance.

For the *Base-initialized* setup, we perform a grid search over learning rates $\{6 \times 10^{-6}, 8 \times 10^{-6}, 1 \times 10^{-5}\}$, inspired by values suggested in ORPO and ASFT, and explore $\lambda \in \{0.1, 0.2, 0.5, 1.0\}$ for 1 and 2 training epochs keeping a similar budget to compare with the *SFT-initialized* setup.

In the *SFT-initialized* setup, we experiment with both $\mathcal{L}_{\text{ORPO}_{\text{Align}}}$ and $\mathcal{L}_{\text{ASFT}_{\text{Align}}}$ alone, as well as in combination with $\mathcal{L}_{\text{SFT}}$, following the original methods. We tune the learning rates $\{5\times10^{-7}, 7\times10^{-7}, 1\times10^{-6}\}$ for one epoch, starting from an SFT model trained for 1 epoch at $6\times10^{-6}$.

## 4.2 $\beta$ SENSITIVITY.

Building on the theoretical insights from Section 3.2, where DAA losses share indistinguishable gradient directions as $\beta \to 0$, we evaluate each method across various $\beta$ values to examine quality-KL trade-offs. In classical DPO, $\beta$ regulates the KL penalty from the reference policy, but setting $\beta$ too small can induce training instability. Therefore, we conduct a thorough sweep of at least six $\beta$ values per DAA, exploring the performance limit of each method. To broaden our analysis, we consider three scenarios:

**Llama 3.2 3B TL;DR.** A relatively simpler Reddit TL;DR summarization task, evaluated via GPT side-by-side comparison on 500 samples from the "golden" validation split Rafailov et al. (2023; 2024).

**Llama 3.2 3B UF.** The UltraChat and UF datasets serve as more challenging alignment settings due to their coverage of diverse and complex tasks, including common sense reasoning, mathematical problem-solving, code generation, logical reasoning, creative writing, and general knowledge.

**Llama 3.1 8B UF.** A larger, more capable model on the same UltraChat and UF datasets, allowing us to assess how increased model capacity influences $\beta$-sensitivity in these diverse tasks.

For the UF-based experiments, we measure model quality primarily using the AlpacaEval 2 Length-Controlled (LC) Win-Rate and ArenaHard (AH) WR, and then track KL divergence from a reference model to construct Pareto fronts. For the TL;DR scenario, we rely on GPT-based preference judgments using 'gpt-4o-2024-08-06' model. Concretely, in each scenario we train models for different values $\beta$, combining them with four possible learning rates $\{1\times10^{-6}, 7\times10^{-7}, 5\times10^{-7}, 3\times10^{-7}\}$. Further implementation details, including training procedures and generation hyperparameters, are provided in Appendix B.

## 4.3 SFT QUALITY.

Although in principle single-stage methods do not require a separate SFT phase, in practice an SFT-trained reference model often improves the final performance of two-stage pipelines (see Section 5.1). Prior work, such as (Zhou et al., 2024), has shown that a small but high-quality dataset can be sufficient for instruction tuning. However, beyond response quality, it remains unclear how the amount of SFT data influences alignment effectiveness. This raises a fundamental question: how much supervised data is actually needed to produce a reference model that yields high-quality results after the subsequent alignment step?

To investigate this, we prepared seven SFT checkpoints by training Llama 3.1 8B Base on 1%, 3%, 5%, 10%, 25%, 50%, and 100% of the UltraChat dataset (2,079, 6,236, 10,393, 20,786, 51,966, 103,932, and 207,865 records, respectively) using our *SFT-initialized* procedure. We then applied each alignment method – using *optimal hyperparameters* from our $\beta$-sensitivity experiments (Appendix Table 8) – to these seven SFT checkpoints and the original base model. Finally, we evaluated all resulting aligned models on AlpacaEval 2 LC, analyzing their performance relative to the fraction of SFT data used.

# 5 RESULTS

## 5.1 RQ1: DOES AN EXPLICIT SFT STAGE IMPROVE THE ALIGNMENT QUALITY OF ORPO AND ASFT?

As shown in Table 1, the performance of ORPO and ASFT methods improves significantly when the alignment loss $\mathcal{L}_{\text{Align}}$ is applied after a preceding SFT stage. In particular, ORPO achieves results comparable to classical DPO in both LC Win Rate and AH WR metrics. In contrast, ASFT shows notable gains in AH WR after the SFT stage, although it still underperforms compared to ORPO or DPO.

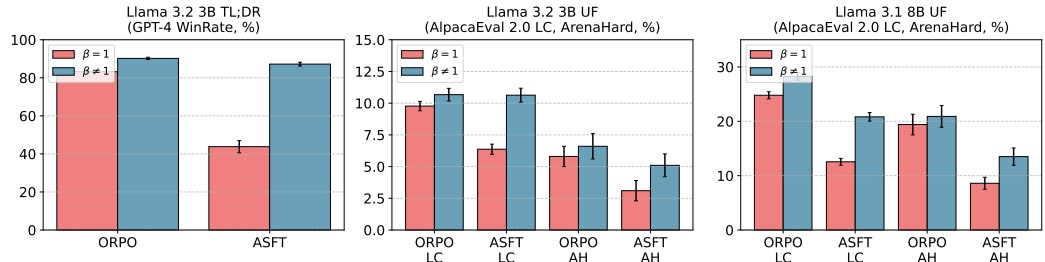

Figure 1: **Impact of the $\beta$ Parameter on ASFT and ORPO Alignment Quality.** The plot shows how tuning $\beta$ (Section 3.1.2) affects both ASFT and ORPO performance. Results are reported for GPT-4 Win Rate in the Llama 3.2 3B TL;DR setup and for AlpacaEval 2 LC Win Rate in the Llama 3.1 8B UF scenario. All other hyperparameters (e.g., learning rates) are selected via grid search, using each method's best configuration at $\beta = 1$ as the baseline. See Section 5.2 for more details.

For single-stage methods, the use of $\lambda = 1$ provides the best results within the explored grid of $\lambda \in \{0.1, 0.2, 0.5, 1.0\}$, especially after two epochs of training. However, combining $\mathcal{L}_{\text{SFT}}$ and $\mathcal{L}_{\text{Align}}$ in a single-stage setup leads to suboptimal results compared to explicitly separating these phases, even when starting from an SFT-trained model. Incorporating an explicit SFT stage improves overall performance for ORPO and ASFT methods. Therefore, all further experiments focus on applying the $\mathcal{L}_{\text{Align}}$ components of ORPO and ASFT on top of an SFT-trained model.

## 5.2 RQ2: Does the tempering factor enhance the alignment quality of ASFT and ORPO?

Figure 1 illustrates that introducing the $\beta$ parameter (as described in Section 3.1.2) improves the performance of both ASFT and ORPO $\mathcal{L}_{\text{Align}}$ in our tested scenarios. For a fair comparison, we used the best-performing learning rate for each baseline — $\mathcal{L}_{\text{ASFT}_{\text{Align}}}$ and $\mathcal{L}_{\text{ORPO}_{\text{Align}}}$ — while fixing $\beta = 1$. In the Llama 3.2 3B TL;DR experiment, these adjustments led to an improvement of +7.0 for ORPO and +43.4 for ASFT in GPT-4 WR. In the Llama 3.1 8B UF setup, tuning $\beta$ provided additional gains of +3.46 for ORPO and +8.27 for ASFT on the AlpacaEval 2 LC WR.

| Init | Method | LC% (std) | WR% (std) | AH% (CI) |
|------|--------|-----------|-----------|----------|
| Base | SFT | 6.7 (0.43) | 4.5 (0.63) | 3.5 (-0.7, 0.8) |
| SFT | ORPO | **24.1** (0.84) | 17.8 (1.17) | 15.3 (-1.6, 1.8) |
| SFT | ASFT | 16.4 (0.72) | 11.9 (0.99) | 10.6 (-1.2, 1.3) |
| Base | ORPO$^\dagger$ | 14.8 (0.71) | 10.3 (0.95) | 8.4 (-1.3, 1.3) |
| Base | ASFT$^\dagger$ | 14.5 (0.73) | 10.2 (0.94) | 7.5 (-1.1, 1.2) |
| SFT | ORPO$^\dagger$ | 13.4 (0.69) | 9.3 (0.91) | 7.7 (-0.9, 1.1) |
| SFT | ASFT$^\dagger$ | 11.4 (0.63) | 7.5 (0.83) | 7.5 (-1.1, 1.1) |
| SFT | DPO | 23.4 (0.85) | **20.0** (1.18) | **17.5** (-1.8, 1.8) |

Table 1: **Base and SFT-initialized alignment methods on the Llama 3.1 8B model with the UF dataset.** SFT-initialized methods demonstrate better performance compared to their traditional formulations without $\mathcal{L}_{\text{SFT}}$. Results marked with † correspond to training with $\mathcal{L}_{\text{SFT}}$, using the best hyperparameters: lr $= 1 \times 10^{-6}$ for ORPO and lr $= 7 \times 10^{-7}$ for ASFT. For other setups, the best hyperparameters are: lr $= 5 \times 10^{-7}$ for standard SFT ORPO/ASFT, and lr $= 1 \times 10^{-5}/6 \times 10^{-6}$ for Base ORPO/ASFT.

## 5.3 RQ3: What factors of DAAs affect alignment quality?

Based on Section 3, we perform a comprehensive evaluation of alignment losses, including DPO, IPO, SimPO, NCA, Cal-DPO, and APO-Zero, as well as enhanced $\mathcal{L}^{\beta}_{\text{ASFT}_{\text{Align}}}$ and $\mathcal{L}^{\beta}_{\text{ORPO}_{\text{Align}}}$ with the introduced parameter $\beta$. Unlike classical methods where $\beta$ typically regulates KL divergence against a reference policy $\pi_{\text{ref}}$, $\beta$ in $\mathcal{L}^{\beta}_{\text{ASFT}_{\text{Align}}}$ and $\mathcal{L}^{\beta}_{\text{ORPO}_{\text{Align}}}$ directly modulates the strength of preference optimization. To explore the upper limits of each method's performance, we performed an extensive hyperparameter search, analyzing both alignment quality and KL divergence. Full implementation details, including training setups and evaluation criteria, are provided in Appendix B.

**Llama 3.2 3B TL;DR:** The comparison of all methods on the Reddit TL;DR validation subset, using their best hyperparameters, shows that most methods achieve a GPT-4 Win Rate exceeding 90%,

indicating robust summarization performance on this relatively straightforward task (see Figure 3 in the Appendix). ASFT is slightly lower at 87.2% Win Rate, but still demonstrates strong overall results.

**Llama 3.2 3B UF and Llama 3.1 8B UF:** Table 3 summarizes the results for both Llama 3.2 3B UF and Llama 3.1 8B UF setups. For the smaller 3B model, the methods perform similarly on LC WR, with slight differences emerging on AH. Although these differences align with the pairwise vs. pointwise distinction (e.g., DPO, IPO, ORPO, SimPO vs. APO-Zero, NCA, Cal-DPO, ASFT), no single approach consistently dominates across metrics. The overlap in confidence intervals further indicates that the results for these methods are statistically similar in this setup, with no clear separation.

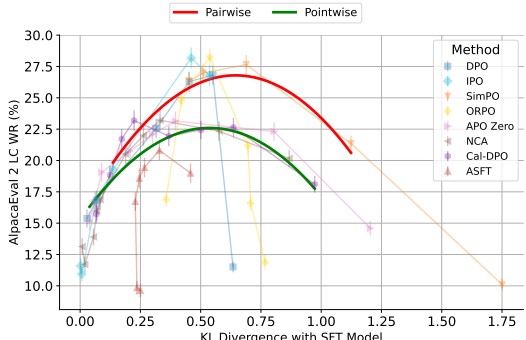

Table 2: **Pareto front for alignment quality and KL divergence.** Results for Llama 3.1 8B UF on AlpacaEval 2 LC. Methods are grouped into pairwise and pointwise categories, with pairwise achieving higher LC values while remaining within overlapping confidence intervals. See Section 5.3 for more details.

In contrast, the 8B model reveals a clearer performance differentiation. Pairwise methods consistently outperformed pointwise ones on AlpacaEval 2 and ArenaHard metrics, with ORPO achieving the highest overall alignment quality. As illustrated in Figure 2, pairwise approaches dominated the KL Pareto front for the larger model, demonstrating their ability to more effectively balance alignment quality and divergence. Pareto fronts for the remaining setups are included in Appendix I for completeness.

| Method | Llama 3.2 3B UF | | | Llama 3.1 8B UF | | |
|---|---|---|---|---|---|---|
| | AlpacaEval 2 | | ArenaHard | AlpacaEval 2 | | ArenaHard |
| | LC% (std) | WR% (std) | WR% (CI) | LC% (std) | WR% (std) | WR% (CI) |
| SFT | 5.02 (0.34) | 3.21 (0.55) | 1.4 (-0.4, 0.4) | 10.27 (0.54) | 5.44 (0.70) | 2.6 (-0.5, 0.6) |
| DPO | **11.43** (0.58) | 11.79 (0.99) | 6.8 (-1.0, 0.9) | 26.82 (0.77) | 23.69 (1.25) | 19.0 (-1.9, 1.8) |
| IPO | 11.24 (0.60) | 11.67 (1.01) | **6.8** (-1.0, 1.1) | 28.18 (0.83) | 24.43 (1.26) | 19.1 (-1.6, 1.5) |
| SimPO | 10.56 (0.44) | 11.94 (0.95) | 6.4 (-1.0, 1.1) | 27.65 (0.77) | 25.62 (1.29) | **21.5** (-1.9, 1.9) |
| ORPO | 10.67 (0.50) | **12.23** (0.97) | 6.6 (-1.0, 1.1) | **28.25** (0.71) | **28.59** (1.33) | 20.9 (-2.0, 2.0) |
| APO Zero | 10.36 (0.53) | 11.22 (0.98) | 6.0 (-1.0, 0.9) | 23.15 (0.76) | 19.03 (1.18) | 17.3 (-1.8, 1.8) |
| NCA | 10.33 (0.53) | 11.02 (0.97) | 5.1 (-0.7, 0.8) | 23.21 (0.80) | 18.67 (1.17) | 15.1 (-1.5, 1.6) |
| Cal-DPO | 10.62 (0.57) | 10.15 (0.94) | 4.8 (-0.9, 0.9) | 23.19 (0.82) | 18.85 (1.18) | 15.2 (-1.5, 1.6) |
| ASFT | 10.63 (0.55) | 9.21 (0.88) | 5.1 (-0.9, 0.9) | 20.82 (0.79) | 16.34 (1.13) | 13.5 (-1.6, 1.5) |

Table 3: **AlpacaEval 2 and ArenaHard Results for Llama 3.2 3B and Llama 3.1 8B UF.** The SFT model was trained on the UltraChat dataset. The best hyperparameters for each method were selected according to Section 4.2. Bold values indicate the best performance for each benchmark, while underlined values represent the second-best performance. See Section 5.3 for more details.

These observations suggest that model capacity plays a significant role in amplifying the advantages of pairwise ranking, where LLMs act as rankers (similar to Liu et al. (2024)). For smaller models, such as the 3B setup, limited capacity may hinder the ability to fully exploit pairwise gradient signals. This hypothesis is supported by additional evidence from the toy example experiment (see Figure 5 in Appendix), where pairwise methods demonstrated performance similar to pointwise methods with weaker MLPs but achieved better ranking accuracy as the model capacity increased. Full details of the toy example setup are provided in Appendix J.

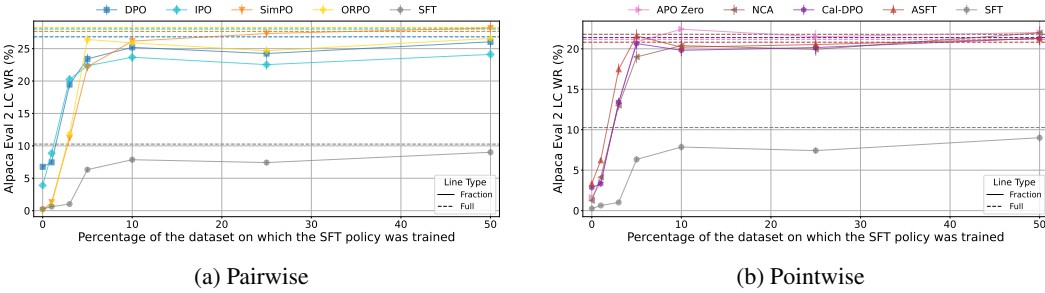

(a) Pairwise                                 (b) Pointwise

Figure 2: **Impact of SFT Dataset Size on Alignment Quality.** Performance of the pairwise (a) and pointwise (b) alignment methods on AlpacaEval 2 (LC WR metric) when the SFT policy is trained on different fractions of the UltraChat dataset. Even a small fraction of SFT data (e.g., 5–10%) yields substantial gains over starting from the raw base model. See Section 5.4 for more details.

### 5.4 RQ4: How does the final alignment quality depend on the amount of data used in the SFT stage?

In Section 5.1, we show that DAAs designed to bypass the SFT phase still underperform compared to models that undergo SFT and are then aligned using a similar preference-optimization loss function *without* the SFT term. As discussed in Section 4.3, this raises the question of how much supervised data is needed to compensate for the additional computation and achieve comparable alignment performance.

To investigate this, we trained seven SFT models on progressively larger UltraChat subsets (1% to 100%) and applied each alignment algorithm to these models and the non-fine-tuned base model, yielding eight initializations per method. Figures 2(a) and 2(b) summarize the results for pairwise and pointwise alignment methods, respectively. As the plots show, no method starting from the raw base model can match the final quality of a method trained with the entire SFT dataset. However, even a modest size expansion of the SFT dataset yields substantial improvements in alignment quality: for example, moving from 3% to 5% of the data more than doubles the AlpacaEval 2 LC score for the final model. Crucially, using only 10% of UltraChat for SFT yields nearly the same quality as using the entire dataset.

Adding an SFT phase requires more overall training, but it *pays off significantly* in the final result. Moreover, one does not need the entire supervised corpus to realize most of these gains; even 5–10% of the data is often enough for DAAs to reach most of their potential.

## 6 Conclusion

This paper presents a comprehensive theoretical and empirical analysis of DAAs. Theoretically, we demonstrated that within each category - odds-based ($r^{\mathrm{odds}}$) and reference-policy-based ($r^{\mathrm{ref}}$) – gradient directions of popular methods align as $\beta \to 0$, revealing shared optimization dynamics within these groups. We also showed that single-stage losses (e.g., ASFT, ORPO) can be extended to two-stage pipelines with an explicit SFT step and optional $\beta$-scaling, enabling greater flexibility. Experimentally, we addressed four core research questions (RQ1–4), exploring single- vs. two-stage training, implicit rewards, objective types, and the impact of the SFT phase. Our key findings are:

- **Include an SFT phase.** An SFT stage consistently improves alignment performance (RQ1), with ORPO achieving +9.3 LC / +6.9 AH and ASFT +1.9 LC / +3.1 AH in the setup from Section 4.1. Even 5–10% of the supervised dataset often suffices to achieve near-optimal results (RQ4).

- **Pairwise methods outperform pointwise objectives.** Alignment quality depends more on the choice between pairwise and pointwise objectives than on the formulation of implicit reward (e.g., $r^{\mathrm{odds}}$ or $r^{\mathrm{ref}}$). Pairwise methods generally perform better (e.g., ORPO outperforming ASFT by +7.43 LC / +7.4 AH in the Llama 3.1 8B UF setup), particularly in larger models (RQ3). Among these, ORPO and SimPO also stand out as practical options for memory-constrained scenarios, as they do not rely on a reference policy.

- **Choose hyperparameters carefully.** Alignment performance is highly sensitive to learning rates and the coefficient $\beta$. We provide optimal configurations for different methods based on comprehensive grid searches in our experimental setups, highlighting the added gains from tuning $\beta$ in odds-based methods, where it controls the strength of preference optimization (RQ2).

**Limitations and Future Work.** Although our study systematically compares DAAs, it has several limitations. We tested a limited set of datasets (UltraChat, UltraFeedback, Reddit TL;DR) and benchmarks (AlpacaEval 2, ArenaHard), which may affect generalizability to other domains. The reliance on GPT-based evaluators can introduce biases. Moreover, we evaluated on 3B–8B models, so the observed advantages of pairwise over pointwise objectives could shift at larger scales.

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

## A  REINFORCEMENT LEARNING FROM HUMAN FEEDBACK

Reinforcement Learning from Human Feedback (RLHF) (Ouyang et al., 2022; Stiennon et al., 2020) is a prominent approach to aligning language models. It generally has three stages:

**Supervised Fine-Tuning (SFT).** During the SFT stage, the model $\pi_\theta$ is trained to follow instructions by maximizing the probability of correct output $y$ given input $x$. For a single training pair $(x, y)$, we define the per-sample SFT loss as $\mathcal{L}_{\text{SFT}}(\pi_\theta, x, y) = -\log \pi_\theta(y \mid x)$. During fine-tuning, we minimize the expectation of this per-sample loss over the training dataset $\mathcal{D}$: $\mathbb{E}_{(x,y) \sim \mathcal{D}}\Big[\mathcal{L}_{\text{SFT}}(\pi_\theta, x, y)\Big]$.

**Reward Modeling (RM).** A reward model $r_\psi(x, y)$ produces a satisfaction score. It is trained on preference pairs using the Bradley-Terry model (Bradley & Terry, 1952): $\mathcal{L}_{\text{RM}}(r_\psi) = -\mathbb{E}_{(x,y_w,y_l) \sim \mathcal{D}}\big[\log \sigma\big(r_\psi(x, y_w) - r_\psi(x, y_l)\big)\big]$, where $y_w$ is the preferred response and $y_l$ is the less preferred one.

**Reward Maximization.** The objective is to generate responses that maximize the learned reward, with a KL penalty to prevent reward hacking: $\max_{\pi_\theta} \mathbb{E}_{x \sim \mathcal{D}, y \sim \pi_\theta(y|x)}\big[r_\phi(x, y)\big] - \beta \mathbb{D}_{\text{KL}}\big[\pi_\theta(x, y) \,\|\, \pi_{\text{ref}}(x, y)\big]$. Reinforcement learning (RL) algorithms are commonly used to optimize this objective (Schulman et al., 2017; Ouyang et al., 2022).

## B  IMPLEMENTATION DETAILS

### B.1  PROBABILITY NORMALIZATION

As discussed in Section 2.1, not all DAAs incorporate length-based probability normalization by default. In this paper, however, we apply such normalization only in cases where it was used in the original methods involving probabilities. This choice avoids introducing extra notation and reduces the cognitive load on the reader. Table 4 summarizes the methods that originally include length-based normalization.

| Method | Use normalization |
|---|---|
| DPO (Rafailov et al., 2023) | ✗ |
| IPO (Azar et al., 2023) | ✗ |
| SimPO (Meng et al., 2024) | ✓ |
| NCA (Chen et al., 2024) | ✗ |
| Cal-DPO (Xiao et al., 2024) | ✗ |
| APO-Zero (D'Oosterlinck et al., 2024) | ✗ |
| ORPO (Hong et al., 2024) | ✓ |
| ASFT (Wang et al., 2024) | ✓ |

Table 4: Methods that include (✓) or omit (✗) length-based probability normalization in their original formulation.

### B.2  TRAINING DETAILS

Our experiments were conducted using the Llama 3.2 3B and Llama 3.1 8B Base models AI@Meta (2024). The training setup, datasets, and hyperparameters were designed to ensure reproducibility and consistency. Unless otherwise noted, the hyperparameters in Table 5 were used across all experiments.

Training was performed on 8 NVIDIA A100 GPUs with 80GB memory each. Depending on the number of epochs, training for each configuration took between 3 to 6 hours.

#### B.2.1  DATASETS.

We used two primary datasets:

| Hyperparameter | Value |
|---|---|
| Max Tokens Length | 1024 (TL;DR setup), 4096 (UF setup) |
| Epochs | 1 *(or 2 when specified)* |
| Learning Rate (SFT) | $6.0 \times 10^{-6}$ |
| Learning Rate (Base Init.) | $\{6.0 \times 10^{-6}, 8.0 \times 10^{-6}, 1.0 \times 10^{-5}\}$ |
| Learning Rate (Alignment) | $\{3.0 \times 10^{-7}, 5.0 \times 10^{-7}, 7.0 \times 10^{-7}, 1.0 \times 10^{-6}\}$ |
| Optimizer | Adam (Kingma & Ba, 2014) |
| Adam $\beta_1$ | 0.9 |
| Adam $\beta_2$ | 0.95 |
| Batch Size | 128 |
| Learning Schedule | Linear Decay |
| Warm-up Ratio | 0.03 |
| Max Gradient Norm | 2 |
| Memory Optimization | DeepSpeed (Rasley et al., 2020) |
| Attention Mechanism | Flash Attention 2 (Dao, 2023) |

Table 5: Representative training hyperparameters for Llama 3.2 3B and Llama 3.1 8B models.

- **Reddit TL;DR** (Bai et al., 2022): used to train the initial SFT model in $\beta$-sensitivity experiments with Llama 3.2 3B model.

- **UltraChat** (Ding et al., 2023): used to train the initial SFT model in $\beta$-sensitivity experiments with Llama 3.2 3B and Llama 3.1 8B models.

- **UltraFeedback** (Cui et al., 2023): used for both SFT (in the *Base vs. SFT-initialized* comparison, where we selected chosen subset from preference pairs) and for pairwise preference optimization in all DAA methods.

The dataset sizes are summarized in Table 6. For *Base* vs. *SFT-initialized* setups, only UltraFeedback was used. For $\beta$-sensitivity experiments, the models were first trained on UltraChat for SFT and subsequently fine-tuned on UltraFeedback. The Reddit TL;DR dataset was processed to remove duplicates, retaining only uniquely preferred summaries for SFT.

| Dataset | Training Examples | Validation Examples |
|---|---|---|
| UltraChat | 207,865 | 23,110 |
| UltraFeedback | 61,135 | 2,000 |
| Reddit TL;DR (SFT) | 41,947 | 11,941 |
| Reddit TL;DR (Preference) | 73,396 | 21,198 |

Table 6: Summary of dataset sizes used for training and validation.

### B.2.2 $\beta$-SENSITIVITY EXPERIMENTS.

We conducted a comprehensive analysis to evaluate the sensitivity of DAA methods to $\beta$, examining its impact on the trade-off between model quality and KL divergence. Each method was trained using six or more distinct $\beta$ values to identify a configuration that achieves stable and effective performance. The specific $\beta$ values tested for each method are as follows:

For each $\beta$, we tested four learning rates ($3.0 \times 10^{-7}, 5.0 \times 10^{-7}, 7.0 \times 10^{-7}, 1.0 \times 10^{-6}$), training on the UltraFeedback dataset. All runs began from an SFT-initialized model trained on UltraChat ($\text{lr} = 6.0 \times 10^{-6}$, 1 epoch). The best-performing learning rate for each $\beta$ was selected to construct Pareto fronts, balancing quality (measured via AlpacaEval 2 LC Win-Rate) and KL divergence.

For SimPO in the Llama 3.1 8B UF setup, the ratio $\frac{\gamma}{\beta} = 0.5$ was kept fixed as recommended by Meng et al. (2024). Additionally, a single learning rate ($\text{lr} = 6.0 \times 10^{-7}$) was tested across all $\beta$ values for this method, as the same datasets and model scale were used. For Llama 3.2 TL;DR and UF setups, we tested four learning rates similar to other DAAs. Beyond the standard $\beta$ values described in Table 7, additional values were explored for specific configurations to reach the extreme

| Method | $\beta$ Values Tested |
|---|---|
| DPO | $\{0.001, 0.003, 0.005, 0.01, 0.05, 0.1\}$ |
| IPO | $\{0.0007, 0.001, 0.005, 0.01, 0.05, 0.1\}$ |
| SimPO | $\{0.05, 0.1, 0.2, 0.5, 1.0, 2.0, 5.0\}$ |
| ORPO | $\{0.05, 0.1, 0.2, 0.5, 1.0, 2.0\}$ |
| ASFT | $\{0.05, 0.1, 0.2, 0.5, 1.0, 2.0\}$ |
| APO-Zero | $\{0.001, 0.003, 0.005, 0.01, 0.05, 0.1, 0.2\}$ |
| Cal-DPO | $\{0.00005, 0.0001, 0.0003, 0.0005, 0.001, 0.003\}$ |
| NCA | $\{0.0001, 0.0003, 0.0005, 0.001, 0.005, 0.007, 0.01, 0.03, 0.05\}$ |

Table 7: Range of $\beta$ values tested for each DAA method on all scenarios.

points of the Pareto front. For example: - $\{0.00001, 0.00003\}$ for Cal-DPO in Llama 3.2 3B TL;DR and UF setups, - $\{0.00001, 0.00003, 0.00005\}$ for NCA in Llama 3.2 3B TL;DR, - $\{0.0003, 0.0005\}$ for APO-Zero in Llama 3.2 3B TL;DR, - $\{0.0003, 0.0005, 0.001, 0.003, 0.005\}$ for ASFT in Llama 3.2 3B TL;DR.

The hyperparameters resulting in the best performance are presented in Table 8.

| Method | Llama 3.2 3B TL;DR | | Llama 3.2 3B UF | | Llama 3.1 8B UF | |
|---|---|---|---|---|---|---|
| | Learning Rate | $\beta$ | Learning Rate | $\beta$ | Learning Rate | $\beta$ |
| DPO | $7.0 \times 10^{-7}$ | 0.05 | $1.0 \times 10^{-6}$ | 0.01 | $1.0 \times 10^{-6}$ | 0.003 |
| IPO | $1.0 \times 10^{-6}$ | 0.005 | $7.0 \times 10^{-7}$ | 0.001 | $1.0 \times 10^{-6}$ | 0.001 |
| SimPO | $3.0 \times 10^{-7}$ | 0.5 | $7.0 \times 10^{-7}$ | 1.0 | $6.0 \times 10^{-7}$ | 1.0 |
| ORPO | $3.0 \times 10^{-7}$ | 0.5 | $5.0 \times 10^{-7}$ | 0.2 | $5.0 \times 10^{-7}$ | 0.5 |
| ASFT | $3.0 \times 10^{-7}$ | 0.001 | $1.0 \times 10^{-6}$ | 0.2 | $7.0 \times 10^{-7}$ | 0.1 |
| APO Zero | $3.0 \times 10^{-7}$ | 0.001 | $3.0 \times 10^{-7}$ | 0.005 | $3.0 \times 10^{-7}$ | 0.003 |
| NCA | $3.0 \times 10^{-7}$ | 0.0001 | $3.0 \times 10^{-7}$ | 0.0005 | $3.0 \times 10^{-7}$ | 0.0003 |
| Cal-DPO | $3.0 \times 10^{-7}$ | 0.00003 | $5.0 \times 10^{-7}$ | 0.0003 | $3.0 \times 10^{-7}$ | 0.0003 |

Table 8: Best hyperparameters for each DAA method across setups.

## B.3 GENERATION DETAILS

We evaluated model performance on AlpacaEval 2 and ArenaHard for UltraFeedback setups, while for the Reddit TL;DR setup, we used side-by-side comparisons with GPT-4o on a curated golden validation subset of 500 samples. Additionally, KL divergence was measured on the validation subset for all setups using the generation hyperparameters listed in Table 9. For ArenaHard, the temperature was set to 0 to adhere to the original benchmark configuration.

| Hyperparameter | Value |
|---|---|
| Temperature | 0.9 |
| Top-k | 40 |
| Top-p | 1.0 |
| Max New Tokens | 256 (TL;DR setup), 4096 (UF setup) |

Table 9: Generation hyperparameters for Llama 3.1 8B and Llama 3.2 3B models.

## C    EQUIVALENCE OF ASFT LOSS AND BINARY CROSS-ENTROPY LOSS

**Lemma C.1.**
$$\log \sigma(r_\theta^{\text{odds}}(y, x)) = \log \pi_\theta(y|x)$$

*Proof.*

$$\log \sigma(r_\theta^{\text{odds}}(y, x)) = \log \sigma(\log \frac{\pi_\theta(y|x)}{1 - \pi_\theta(y|x)}) = \log \frac{1}{1 + e^{\log(1 - \pi_\theta(y|x)) - \log(\pi_\theta(y|x))}}$$

$$= \log \frac{1}{1 + \frac{1 - \pi_\theta(y|x)}{\pi_\theta(y|x)}} = -\log \left(1 + \frac{1 - \pi_\theta(y|x)}{\pi_\theta(y|x)}\right) = -\log \frac{\pi_\theta(y|x) + 1 - \pi_\theta(y|x)}{\pi_\theta(y|x)} = \log \pi_\theta(y|x).$$

$\square$

**Lemma C.2.**
$$\log \sigma(-r_\theta^{\text{odds}}(y, x)) = \log \left(1 - \pi_\theta(y|x)\right)$$

*Proof.*

$$\log \sigma(-r_\theta^{\text{odds}}(y, x)) = \log \sigma(-\log \frac{\pi_\theta(y|x)}{1 - \pi_\theta(y|x)}) =$$

$$\log \frac{1}{1 + e^{\log(\pi_\theta(y|x)) - \log(1 - \pi_\theta(y|x))}} = \log \frac{1}{1 + \frac{\pi_\theta(y|x)}{1 - \pi_\theta(y|x)}} =$$

$$-\log \left(1 + \frac{\pi_\theta(y|x)}{1 - \pi_\theta(y|x)}\right) = -\log \frac{1 - \pi_\theta(y|x) + \pi_\theta(y|x)}{1 - \pi_\theta(y|x)} = \log(1 - \pi_\theta(y|x)).$$

$\square$

**Theorem C.3.** $\mathcal{L}_{\text{ASFT}}$ *is equivalent to the binary cross-entropy loss, encompassing both likelihood and unlikelihood components:*

$$\mathcal{L}_{\text{ASFT}} = -(1 + \lambda) \log \pi_\theta(y_w|x) - \lambda \log \left(1 - \pi_\theta(y_l|x)\right).$$

*Proof.* To show that $\mathcal{L}_{\text{ASFT}}$ is equivalent to the BCE loss, we start with the definition:

$$\mathcal{L}_{\text{ASFT}} = -\log \pi_\theta(y_w|x) - \lambda \log \sigma(r_\theta^{\text{odds}}(y_w, x)) - \lambda \log \sigma(-r_\theta^{\text{odds}}(y_l, x)),$$

where $r_\theta^{\text{odds}}(y, x) = \frac{\pi_\theta(y|x)}{1 - \pi_\theta(y, x)}$. Applying Lemma C.1 and Lemma C.2 to the expression, we obtain:

$$\mathcal{L}_{\text{ASFT}} = -\log \pi_\theta(y_w|x) - \lambda \log \pi_\theta(y_w|x) - \lambda \log \left(1 - \pi_\theta(y_l|x)\right)$$
$$= -(1 + \lambda) \log \pi_\theta(y_w|x) - \lambda \log(1 - \pi_\theta(y_l|x)).$$

$\square$

## D    RELATIONSHIP BETWEEN ORPO AND ASFT LOSS FUNCTIONS

**Theorem D.1.** $\mathcal{L}_{\text{ORPO}}$ *can be expressed as:*

$$\mathcal{L}_{\text{ORPO}} = \mathcal{L}_{\text{ASFT}} + \lambda \log \left(\pi_\theta(y_w|x)(1 - \pi_\theta(y_l|x)) + \pi_\theta(y_l|x)(1 - \pi_\theta(y_w|x))\right).$$

*Proof.* We start by defining the ORPO loss:

$$\mathcal{L}_{\text{ORPO}} = -\log \pi_\theta(y_w|x) - \lambda \log \sigma \left(\log \frac{\pi(y_w|x)}{1 - \pi(y_w|x)} - \log \frac{\pi(y_l|x)}{1 - \pi(y_l|x)}\right).$$

Expanding the second term using the identity $\log \sigma(x) = x - \log(e^x + 1)$, we get:

$$-\log \sigma\left( \log \frac{\pi_\theta(y_w|x)}{1 - \pi_\theta(y_w|x)} - \log \frac{\pi_\theta(y_l|x)}{1 - \pi_\theta(y_l|x)} \right)$$

$$= \log \frac{1 - \pi_\theta(y_w|x)}{\pi_\theta(y_w|x)} + \log \frac{\pi_\theta(y_l|x)}{1 - \pi_\theta(y_l|x)} + \log\left( \frac{\pi_\theta(y_w|x)(1 - \pi_\theta(y_l|x))}{\pi_\theta(y_l|x)(1 - \pi_\theta(y_w|x))} + 1 \right)$$

$$= \log \frac{1 - \pi_\theta(y_w|x)}{\pi_\theta(y_w|x)} + \log \frac{\pi_\theta(y_l|x)}{1 - \pi_\theta(y_l|x)} + \log\left( \frac{\pi_\theta(y_w|x) - 2\pi_\theta(y_w|x)\pi_\theta(y_l|x) + \pi_\theta(y_l|x)}{\pi_\theta(y_l|x)(1 - \pi_\theta(y_w|x))} \right)$$

$$= \underbrace{-\log \pi_\theta(y_w|x) - \log(1 - \pi_\theta(y_l|x)) + \log\big(\pi_\theta(y_w|x) - 2\pi_\theta(y_w|x)\pi_\theta(y_l|x) + \pi_\theta(y_l|x)\big)}_{\text{ORPO}_{\text{Align}}}.$$

Combining all terms, we obtain:

$$\mathcal{L}_{\text{ORPO}} = -(1 + \lambda) \log \pi_\theta(y_w|x) - \lambda \log(1 - \pi_\theta(y_l|x))$$
$$+ \lambda \log \big(\pi_\theta(y_w|x)(1 - \pi_\theta(y_l|x)) + \pi_\theta(y_l|x)(1 - \pi_\theta(y_w|x))\big)$$
$$= \mathcal{L}_{\text{ASFT}} + \lambda \log \big(\pi_\theta(y_w|x)(1 - \pi_\theta(y_l|x)) + \pi_\theta(y_l|x)(1 - \pi_\theta(y_w|x))\big)$$

$\square$

## E  PROOF OF THEOREM 3.3

**Theorem E.1** (Collinearity of $\beta$-ASFT and ORPO Gradients). *Let*

$$\mathcal{L}_{\text{ASFT}_{\text{Align}}}^{\beta} = -\log \sigma\big(\beta\, r_\theta^{\text{odds}}(y_w, x)\big) - \log \sigma\big(-\beta\, r_\theta^{\text{odds}}(y_l, x)\big),$$

*where*

$$r_\theta^{\text{odds}}(y, x) = \log\left( \frac{\pi_\theta(y|x)}{1 - \pi_\theta(y|x)} \right).$$

*Define the ORPO alignment loss as*

$$\mathcal{L}_{\text{ORPO}_{\text{Align}}} = -\log \sigma\big(r_\theta^{\text{odds}}(y_w, x) - r_\theta^{\text{odds}}(y_l, x)\big).$$

*Then,*

$$\lim_{\beta \to 0} \frac{\nabla_\theta \mathcal{L}_{\text{ASFT}_{\text{Align}}}^{\beta}}{\left\| \nabla_\theta \mathcal{L}_{\text{ASFT}_{\text{Align}}}^{\beta} \right\|} = \frac{\nabla_\theta \mathcal{L}_{\text{ORPO}_{\text{Align}}}}{\left\| \nabla_\theta \mathcal{L}_{\text{ORPO}_{\text{Align}}} \right\|},$$

*i.e., their gradients become collinear in the same direction as $\beta \to 0$.*

*Proof.* **Step 1. Gradient of $\beta$-ASFT.**
Denote $p_w = \pi_\theta(y_w \mid x)$, $p_l = \pi_\theta(y_l \mid x)$. Then

$$r_\theta^{\text{odds}}(y_w, x) = \log\left( \frac{p_w}{1 - p_w} \right), \quad r_\theta^{\text{odds}}(y_l, x) = \log\left( \frac{p_l}{1 - p_l} \right).$$

By definition,

$$\mathcal{L}_{\text{ASFT}_{\text{Align}}}^{\beta} = -\log \sigma\big(\beta\, r_\theta^{\text{odds}}(y_w, x)\big) - \log \sigma\big(-\beta\, r_\theta^{\text{odds}}(y_l, x)\big).$$

For small $\beta$, a first-order Taylor expansion of $\sigma(\beta z)$ around $0$ yields $\sigma(\beta z) = \frac{1}{2} + \frac{\beta z}{4} + O(\beta^2)$. Thus, $\sigma(\beta\, r_\theta^{\text{odds}}(y_w, x)) \approx \frac{1}{2}$ and $\sigma(-\beta\, r_\theta^{\text{odds}}(y_l, x)) \approx \frac{1}{2}$. Taking gradients and applying the chain rule gives each term approximately proportional to $\pm\, \beta\, \nabla_\theta[r_\theta^{\text{odds}}(\cdot)]$. Concretely,

$$\nabla_\theta\big[-\log \sigma(\beta\, r_\theta^{\text{odds}}(y_w, x))\big] \approx -\frac{\beta}{2} \nabla_\theta\big[r_\theta^{\text{odds}}(y_w, x)\big],$$

$$\nabla_\theta\big[-\log \sigma(-\beta\, r_\theta^{\text{odds}}(y_l, x))\big] \approx +\frac{\beta}{2} \nabla_\theta\big[r_\theta^{\text{odds}}(y_l, x)\big].$$

Hence, summing up,

$$\nabla_\theta \mathcal{L}_{\text{ASFT}_{\text{Align}}}^{\beta} \approx \frac{\beta}{2}\left[ \nabla_\theta r_\theta^{\text{odds}}(y_l, x) - \nabla_\theta r_\theta^{\text{odds}}(y_w, x) \right].$$

Observe that $\beta > 0$ implies the overall scalar factor $\frac{\beta}{2}$ is strictly *positive* in front of the difference of gradients.

**Step 2. Gradient of ORPO alignment loss.**
Define $\Delta r_\theta^{\mathrm{odds}}(x) = r_\theta^{\mathrm{odds}}(y_w, x) - r_\theta^{\mathrm{odds}}(y_l, x)$. Then

$$\mathcal{L}_{\mathrm{ORPO}_{\mathrm{Align}}} = -\log \sigma\big(\Delta r_\theta^{\mathrm{odds}}(x)\big).$$

Its gradient (using the chain rule) is proportional to

$$\nabla_\theta \mathcal{L}_{\mathrm{ORPO}_{\mathrm{Align}}} \propto -\nabla_\theta\big[r_\theta^{\mathrm{odds}}(y_w, x) - r_\theta^{\mathrm{odds}}(y_l, x)\big] = \nabla_\theta r_\theta^{\mathrm{odds}}(y_l, x) - \nabla_\theta r_\theta^{\mathrm{odds}}(y_w, x).$$

Up to a strictly positive logistic factor (since $\sigma(\cdot) \in (0, 1)$), the coefficient in front of $\nabla_\theta[r_\theta^{\mathrm{odds}}(\cdot)]$ remains negative, but we track the *absolute* scalar to see it is positive. Indeed, one can write

$$-\nabla_\theta\big(\Delta r_\theta^{\mathrm{odds}}(x)\big) = \kappa_{\mathrm{ORPO}} \, \nabla_\theta r_\theta^{\mathrm{odds}}(y_l, x) - \kappa_{\mathrm{ORPO}} \, \nabla_\theta r_\theta^{\mathrm{odds}}(y_w, x), \quad \kappa_{\mathrm{ORPO}} > 0.$$

**Step 3. Conclusion (positive collinearity).**
Comparing the two gradients:

$$\nabla_\theta \mathcal{L}_{\mathrm{ASFT}_{\mathrm{Align}}}^\beta \approx \frac{\beta}{2}\big[\nabla_\theta r_\theta^{\mathrm{odds}}(y_l, x) - \nabla_\theta r_\theta^{\mathrm{odds}}(y_w, x)\big], \quad \nabla_\theta \mathcal{L}_{\mathrm{ORPO}_{\mathrm{Align}}} \propto \big[\nabla_\theta r_\theta^{\mathrm{odds}}(y_l, x) - \nabla_\theta r_\theta^{\mathrm{odds}}(y_w, x)\big].$$

The ratio is thus strictly *positive* for small $\beta$. Consequently,

$$\lim_{\beta \to 0} \frac{\nabla_\theta \mathcal{L}_{\mathrm{ASFT}_{\mathrm{Align}}}^\beta}{\big\|\nabla_\theta \mathcal{L}_{\mathrm{ASFT}_{\mathrm{Align}}}^\beta\big\|} = \frac{\nabla_\theta \mathcal{L}_{\mathrm{ORPO}_{\mathrm{Align}}}}{\big\|\nabla_\theta \mathcal{L}_{\mathrm{ORPO}_{\mathrm{Align}}}\big\|},$$

establishing collinearity in the same direction. $\square$

## F   PROOF OF THEOREM 3.4

**Theorem F.1** (Collinearity of $\beta$-ORPO and ORPO Gradients). *Let*

$$\Delta r_\theta^{\mathrm{odds}}(x) = r_\theta^{\mathrm{odds}}(y_w, x) - r_\theta^{\mathrm{odds}}(y_l, x),$$

*and consider*

$$\mathcal{L}_{\mathrm{ORPO}_{\mathrm{Align}}}^\beta = -\log \sigma\big(\beta \, \Delta r_\theta^{\mathrm{odds}}(x)\big).$$

*Its gradient is collinear with the gradient of the standard ORPO alignment loss*

$$\mathcal{L}_{\mathrm{ORPO}_{\mathrm{Align}}} = -\log \sigma\big(\Delta r_\theta^{\mathrm{odds}}(x)\big)$$

*for any fixed $\beta > 0$. Formally,*

$$\frac{\nabla_\theta \mathcal{L}_{\mathrm{ORPO}_{\mathrm{Align}}}^\beta}{\big\|\nabla_\theta \mathcal{L}_{\mathrm{ORPO}_{\mathrm{Align}}}^\beta\big\|} = \frac{\nabla_\theta \mathcal{L}_{\mathrm{ORPO}_{\mathrm{Align}}}}{\big\|\nabla_\theta \mathcal{L}_{\mathrm{ORPO}_{\mathrm{Align}}}\big\|}.$$

*Proof.* **Step 1. Gradient of $\beta$-ORPO.**
Let $\Delta r_\theta^{\mathrm{odds}}(x) = r_\theta^{\mathrm{odds}}(y_w, x) - r_\theta^{\mathrm{odds}}(y_l, x)$. Then

$$\mathcal{L}_{\mathrm{ORPO}_{\mathrm{Align}}}^\beta = -\log \sigma\big(\beta \, \Delta r_\theta^{\mathrm{odds}}(x)\big).$$

By the chain rule,

$$\nabla_\theta \mathcal{L}_{\mathrm{ORPO}_{\mathrm{Align}}}^\beta = -\frac{1}{\sigma\big(\beta \, \Delta r_\theta^{\mathrm{odds}}(x)\big)} \, \sigma'\big(\beta \, \Delta r_\theta^{\mathrm{odds}}(x)\big) \, \beta \, \nabla_\theta\big[\Delta r_\theta^{\mathrm{odds}}(x)\big].$$

Since $\sigma'(z) = \sigma(z)\,[1 - \sigma(z)]$, we have

$$-\frac{1}{\sigma\big(\beta \, \Delta r_\theta^{\mathrm{odds}}(x)\big)} \, \sigma'\big(\beta \, \Delta r_\theta^{\mathrm{odds}}(x)\big) = -\beta\big[1 - \sigma\big(\beta \, \Delta r_\theta^{\mathrm{odds}}(x)\big)\big].$$

Thus,

$$\nabla_\theta \mathcal{L}^\beta_{\text{ORPO}_{\text{Align}}} = -\beta \Big[ 1 - \sigma\big(\beta \, \Delta r^{\text{odds}}_\theta(x)\big) \Big] \nabla_\theta \big[\Delta r^{\text{odds}}_\theta(x)\big].$$

Since $\beta > 0$ and $1 - \sigma(\cdot) > 0$, the factor multiplying $\nabla_\theta[\Delta r^{\text{odds}}_\theta(x)]$ is strictly *negative*.

**Step 2. Gradient of standard ORPO (i.e. $\beta = 1$).**
For

$$\mathcal{L}_{\text{ORPO}_{\text{Align}}} = -\log \sigma\big(\Delta r^{\text{odds}}_\theta(x)\big),$$

the gradient is

$$\nabla_\theta \mathcal{L}_{\text{ORPO}_{\text{Align}}} = -\big[ 1 - \sigma(\Delta r^{\text{odds}}_\theta(x)) \big] \nabla_\theta \big[\Delta r^{\text{odds}}_\theta(x)\big].$$

This also has a strictly negative scalar in front of $\nabla_\theta\big[\Delta r^{\text{odds}}_\theta(x)\big]$.

**Step 3. Conclusion (exact positive ratio).**
Since $\nabla_\theta \mathcal{L}^\beta_{\text{ORPO}_{\text{Align}}}$ and $\nabla_\theta \mathcal{L}_{\text{ORPO}_{\text{Align}}}$ both differ from $\nabla_\theta\big[\Delta r^{\text{odds}}_\theta(x)\big]$ by a *negative* coefficient, it follows that these two gradients coincide up to a strictly *positive* factor:

$$\nabla_\theta \mathcal{L}^\beta_{\text{ORPO}_{\text{Align}}} = \kappa(\beta) \, \nabla_\theta \mathcal{L}_{\text{ORPO}_{\text{Align}}}, \quad \kappa(\beta) > 0.$$

Hence

$$\frac{\nabla_\theta \mathcal{L}^\beta_{\text{ORPO}_{\text{Align}}}}{\|\nabla_\theta \mathcal{L}^\beta_{\text{ORPO}_{\text{Align}}}\|} = \frac{\nabla_\theta \mathcal{L}_{\text{ORPO}_{\text{Align}}}}{\|\nabla_\theta \mathcal{L}_{\text{ORPO}_{\text{Align}}}\|},$$

proving the claimed collinearity (in the same direction) for every fixed $\beta > 0$. □

## G  PROOF OF THEOREM 3.5

**Theorem G.1** (Unified Collinearity of DPO with IPO, SimPO, NCA, Cal-DPO, and APO-Zero).
*Let*

$$\Delta r^{\text{ref}}_\theta(x) = r^{\text{ref}}_\theta\big(y_w, x\big) - r^{\text{ref}}_\theta\big(y_l, x\big),$$

*and define the DPO loss*

$$\mathcal{L}_{\text{DPO}} = -\log\Big(\sigma\big(\beta \, \Delta r^{\text{ref}}_\theta(x)\big)\Big), \quad \beta > 0.$$

*For each method $X \in \{\text{IPO, SimPO, NCA, Cal-DPO, APO-Zero}\}$, as $\beta \to 0$, the gradient of $\mathcal{L}_X$ is asymptotically collinear (i.e., it differs by a* positive *factor) with the gradient of $\mathcal{L}_{\text{DPO}}$. Formally,*

$$\lim_{\beta \to 0} \frac{\nabla_\theta \mathcal{L}_X}{\|\nabla_\theta \mathcal{L}_X\|} = \frac{\nabla_\theta \mathcal{L}_{\text{DPO}}}{\|\nabla_\theta \mathcal{L}_{\text{DPO}}\|}.$$

*Proof of Theorem 3.5.* **Step 1: DPO as the baseline (tracking its sign).**
By definition,

$$\mathcal{L}_{\text{DPO}} = -\log \sigma\big(\beta \, \Delta r^{\text{ref}}_\theta(x)\big).$$

Since $\sigma(u) = 1/(1 + e^{-u})$, for $\beta > 0$, one computes

$$\nabla_\theta \mathcal{L}_{\text{DPO}} = -\beta \Big[ 1 - \sigma\big(\beta \, \Delta r^{\text{ref}}_\theta(x)\big) \Big] \nabla_\theta \, \Delta r^{\text{ref}}_\theta(x).$$

Observe that $\beta > 0$ and $\sigma(\cdot) \in (0, 1)$ imply

$$1 - \sigma\big(\beta \, \Delta r^{\text{ref}}_\theta(x)\big) > 0.$$

Hence the factor multiplying $\nabla_\theta \, \Delta r^{\text{ref}}_\theta(x)$ is *negative*. To unify directions by a *positive* multiple, note

$$-\nabla_\theta \mathcal{L}_{\text{DPO}} = \beta \Big[ 1 - \sigma\big(\beta \, \Delta r^{\text{ref}}_\theta(x)\big) \Big] \nabla_\theta \, \Delta r^{\text{ref}}_\theta(x),$$

which has a strictly positive scalar in front. Thus, $\nabla_\theta \mathcal{L}_{\text{DPO}}$ is collinear with $\nabla_\theta \, \Delta r^{\text{ref}}_\theta$, and in particular its *negative* is a positive multiple of $\nabla_\theta \, \Delta r^{\text{ref}}_\theta$.

**Step 2: IPO.**
The IPO loss is

$$\mathcal{L}_{\text{IPO}} = \left(\Delta r_\theta^{\text{ref}}(x) - \tfrac{1}{2\beta}\right)^2.$$

Its gradient is

$$\nabla_\theta \mathcal{L}_{\text{IPO}} = 2\left(\Delta r_\theta^{\text{ref}}(x) - \tfrac{1}{2\beta}\right)\nabla_\theta \Delta r_\theta^{\text{ref}}(x).$$

As $\beta \to 0$, the term $\tfrac{1}{2\beta}$ dominates $\Delta r_\theta^{\text{ref}}(x)$. Hence,

$$\Delta r_\theta^{\text{ref}}(x) - \tfrac{1}{2\beta} \approx -\tfrac{1}{2\beta},$$

so

$$\nabla_\theta \mathcal{L}_{\text{IPO}} \approx -\frac{1}{\beta}\nabla_\theta \Delta r_\theta^{\text{ref}}(x).$$

We compare this with

$$\nabla_\theta \mathcal{L}_{\text{DPO}} = -\beta\left[1 - \sigma\left(\beta\,\Delta r_\theta^{\text{ref}}(x)\right)\right]\nabla_\theta \Delta r_\theta^{\text{ref}}(x).$$

Both gradients are negative multiples of $\nabla_\theta \Delta r_\theta^{\text{ref}}(x)$. Therefore,

$$\nabla_\theta \mathcal{L}_{\text{IPO}} = \kappa_{\text{IPO}}(\beta)\,\nabla_\theta \mathcal{L}_{\text{DPO}}, \quad \text{with } \kappa_{\text{IPO}}(\beta) > 0 \text{ as } \beta \to 0.$$

Hence they are collinear in the *same* direction asymptotically.

**Step 3: SimPO.**
The SimPO loss is

$$\mathcal{L}_{\text{SimPO}} = -\log\sigma\left(\beta\,\Delta s_\theta - \gamma\right),$$

where $\Delta s_\theta = \log\pi_\theta(y_w \mid x) - \log\pi_\theta(y_l \mid x)$. Its gradient takes the form

$$\nabla_\theta \mathcal{L}_{\text{SimPO}} = -\frac{\beta\left[1 - \sigma(\beta\,\Delta s_\theta - \gamma)\right]}{\sigma(\beta\,\Delta s_\theta - \gamma)}\,\nabla_\theta \Delta s_\theta.$$

Again, $\beta > 0$ and $1 - \sigma(\cdot) > 0$. Also, $\sigma(\beta\,\Delta s_\theta - \gamma) \in (0,1)$. Thus the prefactor

$$-\frac{\beta\left[1 - \sigma(\beta\,\Delta s_\theta - \gamma)\right]}{\sigma(\beta\,\Delta s_\theta - \gamma)}$$

is strictly negative for each $\beta > 0$. Therefore, just like DPO, $\nabla_\theta \mathcal{L}_{\text{SimPO}}$ is in the negative direction of $\nabla_\theta \Delta s_\theta$. But $\nabla_\theta \Delta s_\theta$ is proportionally the same as $\nabla_\theta \Delta r_\theta^{\text{ref}}$ for small-$\beta$ expansions (both are differences of log-likelihood or reward-like terms). So

$$\nabla_\theta \mathcal{L}_{\text{SimPO}} = \kappa_{\text{SimPO}}(\beta)\,\nabla_\theta \mathcal{L}_{\text{DPO}}, \quad \kappa_{\text{SimPO}}(\beta) > 0 \text{ for small } \beta.$$

Hence they are collinear with a positive factor in the low-$\beta$ limit.

**Step 4: NCA.**
Define

$$r_w^{\text{ref}} = r_\theta^{\text{ref}}(y_w, x), \quad r_l^{\text{ref}} = r_\theta^{\text{ref}}(y_l, x).$$

Then NCA is

$$\mathcal{L}_{\text{NCA}} = -\log\sigma\left(\beta\,r_w^{\text{ref}}\right) - \tfrac{1}{2}\log\sigma\left(-\beta\,r_w^{\text{ref}}\right) - \tfrac{1}{2}\log\sigma\left(-\beta\,r_l^{\text{ref}}\right).$$

For small $\beta$, expand

$$\sigma(\beta\,z) = \frac{1}{2} + \frac{\beta\,z}{4} + O(\beta^2),$$

so $\log\sigma(\beta\,z) = \log\tfrac{1}{2} + \log\left(1 + \tfrac{\beta\,z}{2} + O(\beta^2)\right)$. Each gradient term then yields a linear-in-$\beta$ combination of $\nabla_\theta\,r_w^{\text{ref}}$ and $\nabla_\theta\,r_l^{\text{ref}}$. Collecting terms shows that, as $\beta \to 0$,

$$\nabla_\theta \mathcal{L}_{\text{NCA}} \propto \beta\,\nabla_\theta\left(r_w^{\text{ref}} - r_l^{\text{ref}}\right) = \beta\,\nabla_\theta \Delta r_\theta^{\text{ref}}(x).$$

Comparing this with $\nabla_\theta \mathcal{L}_{\mathrm{DPO}} = -\beta\big[1 - \sigma(\dots)\big]\nabla_\theta \Delta r_\theta^{\mathrm{ref}}(x)$ reveals another negative factor on the DPO side. In ratio form,

$$\nabla_\theta \mathcal{L}_{\mathrm{NCA}} = \kappa_{\mathrm{NCA}}(\beta)\,\nabla_\theta \mathcal{L}_{\mathrm{DPO}} \quad \text{with } \kappa_{\mathrm{NCA}}(\beta) > 0 \text{ for small } \beta.$$

Hence collinearity follows.

**Step 5: Cal-DPO.**
The Cal-DPO loss is

$$\mathcal{L}_{\mathrm{Cal\text{-}DPO}} = -\log\sigma\big(\Delta r_\theta^{\mathrm{ref}}(x)\big) + \big(r_w^{\mathrm{ref}} - \tfrac{1}{2\beta}\big)^2 + \big(r_l^{\mathrm{ref}} + \tfrac{1}{2\beta}\big)^2.$$

For $\beta$ near 0, the large constants $\pm\frac{1}{2\beta}$ dominate. The gradient w.r.t. $\theta$ in these squared terms is effectively

$$\propto -\tfrac{1}{\beta}\,\nabla_\theta\,r_w^{\mathrm{ref}} \;+\; \tfrac{1}{\beta}\,\nabla_\theta\,r_l^{\mathrm{ref}} \;=\; -\tfrac{1}{\beta}\,\nabla_\theta\big(r_w^{\mathrm{ref}} - r_l^{\mathrm{ref}}\big) \;=\; -\tfrac{1}{\beta}\,\nabla_\theta\,\Delta r_\theta^{\mathrm{ref}}(x).$$

Since $\nabla_\theta\,\mathcal{L}_{\mathrm{DPO}}$ has the same negative sign structure in front of $\nabla_\theta\,\Delta r_\theta^{\mathrm{ref}}$, their ratio is again positive. Thus

$$\nabla_\theta\,\mathcal{L}_{\mathrm{Cal\text{-}DPO}} = \kappa_{\mathrm{Cal\text{-}DPO}}(\beta)\,\nabla_\theta\,\mathcal{L}_{\mathrm{DPO}} \quad \text{with } \kappa_{\mathrm{Cal\text{-}DPO}}(\beta) > 0 \text{ as } \beta \to 0.$$

**Step 6: APO-Zero.**
APO-Zero is given by

$$\mathcal{L}_{\mathrm{APO\text{-}Zero}} = -\sigma\big(\beta\,r_w^{\mathrm{ref}}\big) \;+\; \sigma\big(\beta\,r_l^{\mathrm{ref}}\big).$$

Its gradient involves terms $\nabla_\theta\,\sigma(\beta\,r_w^{\mathrm{ref}})$ and $\nabla_\theta\,\sigma(\beta\,r_l^{\mathrm{ref}})$, each proportional to $\beta\,\nabla_\theta\,r_w^{\mathrm{ref}}$ and $\beta\,\nabla_\theta\,r_l^{\mathrm{ref}}$. Subtracting these yields

$$\nabla_\theta\,\mathcal{L}_{\mathrm{APO\text{-}Zero}} \;\propto\; -\beta\,\nabla_\theta\big(r_w^{\mathrm{ref}} - r_l^{\mathrm{ref}}\big) \;=\; -\beta\,\nabla_\theta\,\Delta r_\theta^{\mathrm{ref}}(x).$$

Since $\nabla_\theta\,\mathcal{L}_{\mathrm{DPO}}$ also has a negative constant factor, their ratio has a positive limit. Therefore,

$$\nabla_\theta\,\mathcal{L}_{\mathrm{APO\text{-}Zero}} = \kappa_{\mathrm{APO\text{-}Zero}}(\beta)\,\nabla_\theta\,\mathcal{L}_{\mathrm{DPO}}, \quad \kappa_{\mathrm{APO\text{-}Zero}}(\beta) > 0 \text{ for small } \beta.$$

**Conclusion.**
In each method $X$, one sees that $\nabla_\theta\,\mathcal{L}_X$ has the same *negative-sign* structure around $\nabla_\theta\,\Delta r_\theta^{\mathrm{ref}}(x)$ as does $\nabla_\theta\,\mathcal{L}_{\mathrm{DPO}}$, ensuring a positive ratio in the limit. Formally,

$$\nabla_\theta\,\mathcal{L}_X = \kappa_X(\beta)\,\nabla_\theta\,\mathcal{L}_{\mathrm{DPO}}, \quad \kappa_X(\beta) > 0, \quad \text{as } \beta \to 0.$$

Thus,

$$\lim_{\beta \to 0} \frac{\nabla_\theta\,\mathcal{L}_X}{\|\nabla_\theta\,\mathcal{L}_X\|} = \frac{\nabla_\theta\,\mathcal{L}_{\mathrm{DPO}}}{\|\nabla_\theta\,\mathcal{L}_{\mathrm{DPO}}\|},$$

which completes the proof of their alignment in the same direction. $\qquad\square$

## H  LLAMA 3.2 3B TL;DR GPT-4 EVALUATION RESULTS

Figure 3 presents a comparison of all methods on the Reddit TL;DR validation subset, using their best hyperparameters.

## I  PARETO FRONTS FOR LLAMA 3.2 SETUPS

The results presented in this section correspond to the best hyperparameter configurations identified during the hyperparameter search described in Section 4.2, including the optimal learning rate for each method. This ensures that the Pareto fronts reflect the upper performance limits for alignment quality.

| | Win / Tie / Lose Rate % |
|---|---|
| SFT | 35.6 / 4.8 / **59.6** |
| DPO | **91.2** / 1.0 / 7.8 |
| IPO | **91.4** / 0.4 / 8.2 |
| SimPO | **91.6** / 0.2 / 8.2 |
| ORPO | **90.2** / 0.6 / 9.2 |
| APO Zero | **92.6** / 0.6 / 6.8 |
| NCA | **91.8** / 1.0 / 7.2 |
| Cal-DPO | **91.4** / 0.4 / 8.2 |
| ASFT | **87.2** / 1.0 / 11.8 |

Figure 3: **GPT-4 Evaluation of Llama 3.2 3B TL;DR setup.** The comparison shows multiple alignment methods (rows) using their best hyperparameters, where each approach aims to generate concise and accurate summaries. Most methods exceed 90% Win Rate; ASFT achieves 87.2%, maintaining robust summarization performance. See Section 5.3 for more details.

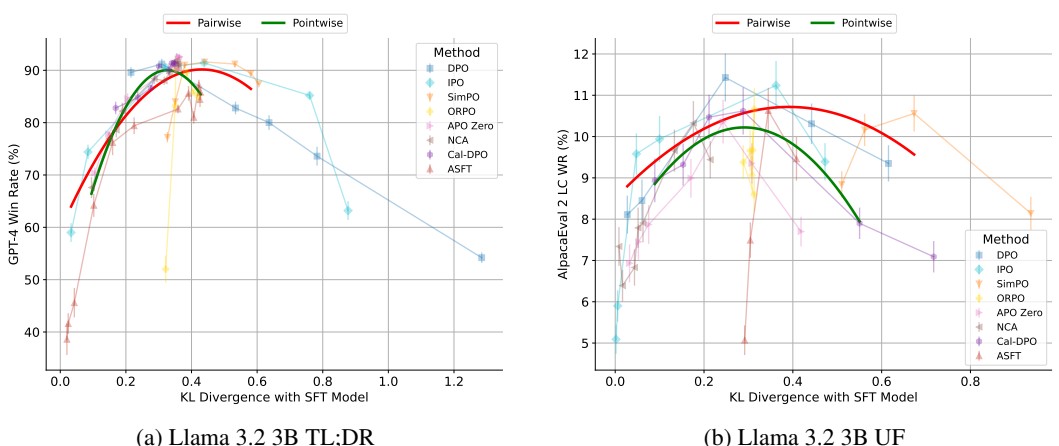

(a) Llama 3.2 3B TL;DR                (b) Llama 3.2 3B UF

Figure 4: **Pareto front for alignment quality and KL divergence.** Results for Llama 3.2 3B TL;DR and UF setups on GPT-4 Win Rate vs. "golden" validation subset and AlpacaEval 2 LC respectively with different $\beta$ values. Methods are grouped into pairwise and pointwise categories. For the summarization task (Llama 3.2 3B TL;DR), both pointwise and pairwise methods achieve strong overall results. For the UF setup, methods also perform similarly within overlapping confidence intervals, indicating no clear separation.

## J   TOY EXAMPLE DETAILS

To analyze the differences between pairwise and pointwise ranking methods, especially with respect to the ranking nature of alignment losses in LLMs, a simplified toy experiment was conducted under a controlled setup. A dataset of 2000 triplets $(x, y_w, y_l)$ was generated, where $x$, $y_w$, and $y_l$ are real-valued scalars satisfying $y_w > y_l$. The data was split into 80% for training and 20% for testing. When the model processes a scalar input $x$ together with a candidate $y$, these two numbers form a vector in $\mathbb{R}^2$, which serves as the input of the Multi-Layer Perceptron (MLP) to predict the reward $r$.

A single-hidden-layer MLP with ReLU activation was used in two capacity settings: lower (hidden size = 1) and higher (hidden size = 3). The model takes $x$ and a candidate $y$ as input, producing a reward $r$ analogous to training a reward model for RLHF Stiennon et al. (2020).

Two losses were evaluated: the pairwise Bradley-Terry loss Bradley & Terry (1952),

$$\mathcal{L}_{\text{Pairwise}} = -\log\big(\sigma(\beta(r_w - r_l))\big),$$

and the pointwise loss,

$$\mathcal{L}_{\text{Pointwise}} = -\big[\log\big(\sigma(\beta r_w)\big) + \log\big(\sigma(-\beta r_l)\big)\big].$$

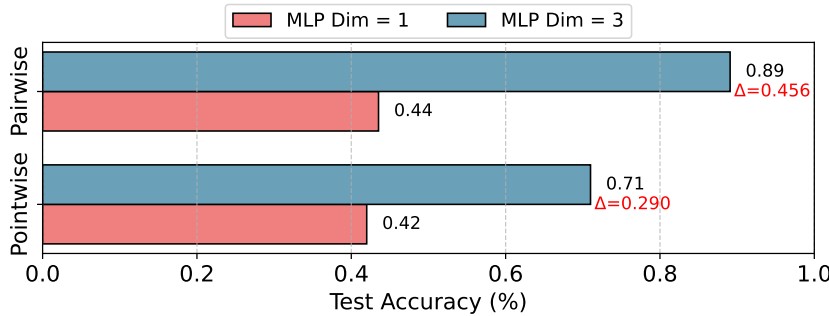

Figure 5: **Pairwise vs. Pointwise Ranking Methods on Toy Example.** Model capacity impacts ranking accuracy, with pairwise methods outperforming pointwise ones as capacity increases. This behavior is consistent with results observed in Llama experiments on the UF dataset. See Section 5.3 for more details.

Each configuration was trained over 100 runs, tuning the learning rate from $\{0.5, 0.3, 0.1, 0.01, 0.03, 0.05\}$ and $\beta$ from $\{5.0, 2.0, 1.0, 0.2, 0.1, 0.05, 0.01\}$. Alignment accuracy was defined as the proportion of cases with $r_w > r_l$.

The results show that both methods yield comparable performance in the low-capacity regime, while pairwise ranking achieves higher accuracy as model capacity increases, mirroring the effects observed in larger-scale experiments from the Section 5.3.

## K    GPT-4 SIDE-BY-SIDE EVALUATION PROMPT

For our Side-By-Side evaluations with `GPT-4o`, we designed a prompt tailored to the Reddit TL;DR dataset to assess *accuracy*, *completeness*, *relevance*, and *conciseness*. The full prompt used in our experiments is detailed below.

```
Act as an impartial judge and evaluate the quality of the summaries provided
by two AI assistants for the text displayed below. Your evaluation should
consider accuracy, completeness, relevance, and conciseness.

You will be given a text, Assistant A's summary, and Assistant B's summary.
Your job is to evaluate which assistant's summary is better based on the
text provided.

Begin your evaluation by comparing both assistants' summaries with the
original text. Identify and correct any inaccuracies.
Ensure the summaries are complete, capturing all essential information
from the text without introducing fabricated details.
Assess the relevance of the information each assistant chose to include
in their summary, ensuring it reflects the core message of the text.
Evaluate the conciseness of the summaries, favoring those that efficiently
convey the necessary information without unnecessary verbosity.
Avoid any position biases and ensure the order in which the summaries
were presented does not influence your decision.
Do not allow the length of the summaries to influence your evaluation,
except in the context of conciseness and efficiency.
Do not favor certain names of the assistants.
Be as objective as possible.
You should only evaluate the summaries provided by both assistants
and NOT the original text itself.
If both summaries are irrelevant, contain hallucinations, or are
inconsistent with the original text, mark the comparison as inconclusive
```

and choose option "C".

After providing your explanation, output your final verdict by strictly following this format:

```
"""
Comparison: <One-sentence comparison>
Winner: <A if assistant A is better, B if assistant B is better, and C for a tie.>
"""
```

