# OpenReview forum: "The Differences Between Direct Alignment Algorithms are a Blur"
_ICLR.cc/2025/Workshop/BuildingTrust — BuildingTrust_

### Official Review · Reviewer_9QPC · 2025-02-22
**Reviews from 9QPC**

**Rating:** 6
**Confidence:** 3

**Review:**

The paper presents a comparative analysis of Direct Alignment Algorithms (DAAs) for aligning large language models (LLMs) with human preferences, bypassing traditional reinforcement learning (RL) and reward modeling steps used in Reinforcement Learning from Human Feedback (RLHF). It categorizes DAAs by ranking losses (pairwise vs. pointwise), implicit reward types (e.g., likelihood ratios vs. odds ratios), and the presence of a Supervised Fine-Tuning (SFT) phase (one-stage vs. two-stage). The authors enhance one-stage methods like ORPO and ASFT by incorporating an explicit SFT phase and a scaling parameter β, demonstrating improved performance. Through theoretical proofs and empirical evaluations, they argue that the distinction between pairwise and pointwise objectives is more critical to alignment quality than the choice of reward formulation, and that even a small SFT dataset can significantly boost performance.

Pros:
- Improves one-stage methods (ORPO, ASFT) with SFT and β, matching two-stage method performance.
- Demonstrates pairwise objectives outperform pointwise objectives, guiding DAA design.
- Shows 5-10% of SFT data achieves near-optimal alignment, lowering computational costs.

Cons:
- The writing could be improved, as there cover too many technical details in the introduction, the authors may consider to smooth their language to make sure readers with limited knowledge of this area can easily follow the paper.
- This reliance on a single automated evaluator (gpt-4o) risks skewing the findings, as the results may not reflect genuine improvements in alignment quality. A more reliable evaluation would include human judgments or a mix of automated metrics to ensure fairness and reduce bias, making the conclusions less convincing without such validation.
- The paper might want to demonstrate some cases or add some instructions regarding the relevance to the topic of this workshop. As the topic seems to be somehow relevant to the workshop as alignment could be useful for LLM trustfulness, but the relationship is not that clear in this paper which requires further clarification, otherwise, it may be considered out-of-scope for this workshop's target.

---

### Official Review · Reviewer_sHeH · 2025-03-02
**This paper studies DAAs and shows their relationships and comparative advantages by identifying key factors and their influence on LLM alignment.**

**Rating:** 6
**Confidence:** 3

**Review:**

This paper studies direct alignment algorithms (DAAs), which directly align large language models (LLMs) without explicit reward modeling and then use reinforcement learning for LLM alignment. The authors categorize various DAAs by their use of loss function (pairwise vs. pointwise), reward function (likelihood ratios, odds ratios), and the need for a supervised fine-tuning (SFT) phase (one-stage vs. two-stage). Further, they have shown the relationship between different direct alignment algorithms. Through experiments on Llama models (3B- 8B) and alignment datasets, authors have shown that (1) two-stage methods outperform one-stage methods, such as ORPO and ASFT, (2) pairwise methods outperform pointwise methods, and (3) using a smaller, high-quality subset of the full SFT dataset can be sufficient to achieve the same performance.

The following are the pros of the paper:

1.⁠ ⁠The authors first provide theoretical results about DAAs, which are then empirically corroborated using established datasets and models.

2.⁠ ⁠As mentioned above, the paper provides practical recommendations for optimizing LLM alignment.

3.⁠ ⁠The paper is overall well-written and easy to understand.

The following are the pros of the paper:

1.⁠ ⁠The experiments only used a specific LLama model (e.g., 3B), set of datasets, and benchmarks, which may limit the real-life applications of the results, especially when larger models are used.

2.⁠ ⁠Using GPT-based preference judgments may introduce potential biases.


This paper studies DAAs and shows their relationships and comparative advantages by identifying key factors and their influence on performance. The results are indeed the first step towards demystifying the "blur" between different DAAs and provide valuable guidance to improve the alignment of language models using human preference feedback.

---

### Official Review · Reviewer_hSxb · 2025-03-03
**This is a good paper providing both theoretical analysis and empirical evidence to answer a few important research questions in the area of Direct Alignment Algorithms.**

**Rating:** 8
**Confidence:** 4

**Review:**

This paper offers a unified analysis of Direct Alignment Algorithms for language model alignment, combining theoretical and empirical approaches. The authors make a contribution by systematically evaluating one-stage versus two-stage methods, providing clear evidence that incorporating an explicit SFT phase substantially improves performance for methods like ORPO and ASFT.

The introduction of the $\beta$ parameter as a way to control preference optimization strength is particularly insightful, demonstrating how seemingly different methods can be understood within a common framework. Their findings that pairwise methods generally outperform pointwise objectives offers practical guidance for practitioners, while simplifying the decision-making process by showing that this choice is more influential than the specific implicit reward formulation.

Despite these strengths, the paper has some limitations. The analysis excludes some competitive algorithms like EXO, which would have provided a more comprehensive picture of the DAA landscape. The work could also benefit from deeper theoretical insights into why pairwise methods consistently outperform pointwise ones.

Overall, this work provides actionable guidance for practitioners implementing alignment methods and establishes a useful framework for understanding and improving DAAs. The paper's systematic approach to addressing key research questions makes it a valuable contribution to the field of language model alignment.

---

### Decision · Program_Chairs · 2025-03-04

Accept